# Learning through ferroelectric domain dynamics in solid-state synapses

Sören Boyn[1,†], Julie Grollier[1], Gwendal Lecerf[2], Bin Xu[3], Nicolas Locatelli[4], Stéphane Fusil[1], Stéphanie Girod[1,†], Cécile Carrétéro[1], Karin Garcia[1], Stéphane Xavier[5], Jean Tomas[2], Laurent Bellaiche[3], Manuel Bibes[1], Agnès Barthélémy[1], Sylvain Saïghi[2] & Vincent Garcia[1]

In the brain, learning is achieved through the ability of synapses to reconfigure the strength by which they connect neurons (synaptic plasticity). In promising solid-state synapses called memristors, conductance can be finely tuned by voltage pulses and set to evolve according to a biological learning rule called spike-timing-dependent plasticity (STDP). Future neuromorphic architectures will comprise billions of such nanosynapses, which require a clear understanding of the physical mechanisms responsible for plasticity. Here we report on synapses based on ferroelectric tunnel junctions and show that STDP can be harnessed from inhomogeneous polarization switching. Through combined scanning probe imaging, electrical transport and atomic-scale molecular dynamics, we demonstrate that conductance variations can be modelled by the nucleation-dominated reversal of domains. Based on this physical model, our simulations show that arrays of ferroelectric nanosynapses can autonomously learn to recognize patterns in a predictable way, opening the path towards unsupervised learning in spiking neural networks.

[1] Unité Mixte de Physique, CNRS, Thales, Univ. Paris Sud, Université Paris-Saclay, Palaiseau 91767, France. [2] University of Bordeaux, IMS, UMR 5218, Talence F-33405, France. [3] Department of Physics and Institute for Nanoscience and Engineering, University of Arkansas Fayetteville, Arkansas 72701, USA. [4] Centre de Nanosciences et de Nanotechnologies, CNRS, Univ. Paris Sud, Université Paris-Saclay, C2N—Orsay, Orsay cedex 91405, France. [5] Thales Research and Technology, 1 Avenue Augustin Fresnel, Campus de l'Ecole Polytechnique, Palaiseau 91767, France. † Present addresses: Electrochemical Materials, ETH Zurich, 8092 Zurich, Switzerland (S.B.); Materials Research and Technology Department, Luxembourg Institute of Science and Technology (LIST), 41 rue du Brill, L-4422 Belvaux, Luxembourg (S.G.). Correspondence and requests for materials should be addressed to S.B. (email: soeren.boyn@gmail.com) or to V.G. (email: vincent.garcia@cnrs-thales.fr).

Cortical information flows from neuron to neuron through synapses of variable connection strength. The overall distribution of the synaptic strengths provides the neural network with memory, while learning is achieved through the synapses' reconfiguration (that is, plasticity)[1]. Several mechanisms regulating the evolution of the synaptic strengths have been proposed[2]. A particularly promising one is spike-timing-dependent plasticity[3] (STDP) through which the synaptic strengths evolve depending on the timing and causality of electrical signals from neighbouring neurons[4] (sketch in Fig. 1a). As observed in biological systems[5], STDP enables learning without any external control on the synaptic strengths or any previous knowledge of the information to be processed. This makes STDP the basis for autonomous, unsupervised learning[6]. The implementation of STDP in artificial neural networks thus emerges as a crucial milestone towards the realization of self-adaptive electronic architectures.

An electronic equivalent of the synapse for artificial neural networks is the memristor[7], a nanoscale device whose resistance depends on the history of electrical signals it was previously subjected to (ref. 8). Memristors thus exhibit plasticity and their conductance can emulate synaptic strength, so that a low resistance corresponds to a strong synaptic connection and a high resistance corresponds to a weak synaptic connection, respectively. Most memristors operate through the motion of ions or atoms in binary oxides ($TiO_2$ (refs 9,10), $Ta_2O_5$ (ref. 11) and so on), Ag-Si/Ag-S nanocomposites[12] or phase change materials[13]. Recently, STDP was demonstrated in individual devices based on such memristor technologies, confirming the potential of memristors for autonomous learning[13–20]. However, the connection between the STDP process and the physical mechanisms underlying resistance changes in these memristors is unclear[7]. In addition, continual variations of the memristor conductance are required for learning new features under an incessant information flow. Here we show that resorting to purely electronic memristors with a high endurance, operating on well-established physical principles is a decisive asset for the future implementation of unsupervised learning[21] in high-density memristive crossbar arrays[22].

## Results

**Ferroelectric synapses.** We work with purely electronic memristors based on ferroelectric tunnel junctions (FTJs), in which an ultrathin ferroelectric film is sandwiched between two electrodes[23,24]. In such devices, sketched in Fig. 1b, the junction resistance sensitively depends on the relative fraction of ferroelectric domains with polarization pointing towards one or the other electrode[25]. Applying voltage pulses to the device modifies the domain population, thereby inducing resistance changes[25]. Furthermore, supertetragonal $BiFeO_3$ (BFO) tunnel barriers combined with $(Ca,Ce)MnO_3$ (CCMO) bottom and Co top electrodes give rise to OFF/ON resistance ratios up to $10^4$ (ref. 26) paired with high endurance and operation speed[27].

Figure 1c shows the dependence of the junction resistance with the amplitude of 100 ns voltage pulses in a Co/BFO/CCMO junction (Methods). In this hysteresis cycle, one clearly notes the existence of voltage thresholds $V_{th}^+$ $(V_{th}^-)$ beyond which switching between low- and high- (high- and low-) resistance states occurs. The existence of such well-defined voltage thresholds (associated with the coercivity of the ferroelectric) makes it possible to implement STDP[28] in these FTJs. According to STDP, if the pre-neuron spikes just before the post-neuron—indicating a 'causal' relationship—the synapse should be strengthened whereas if the pre-neuron spikes just after the post-neuron—indicating an 'anticausal' relationship—the synapse should be

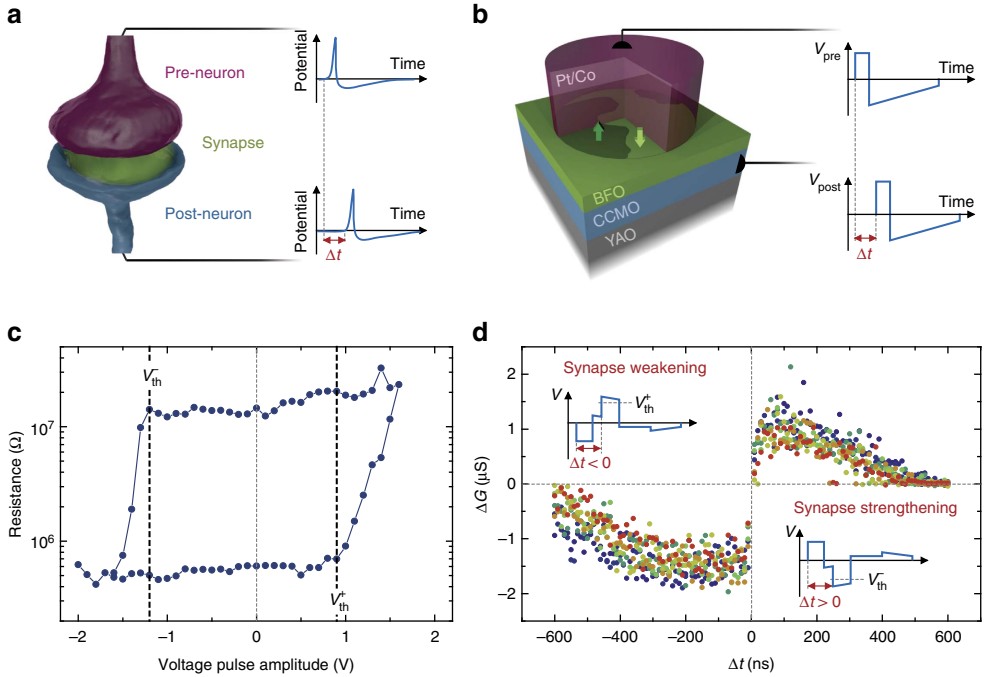

**Figure 1 | Artificial synapses based on FTJs.** (**a**) Sketch of pre- and post-neurons connected by a synapse. The synaptic transmission is modulated by the causality ($\Delta t$) of neuron spikes. (**b**) Sketch of the ferroelectric memristor where a ferroelectric tunnel barrier of $BiFeO_3$ (BFO) is sandwiched between a bottom electrode of $(Ca,Ce)MnO_3$ (CCMO) and a top submicron pillar of Pt/Co. YAO stands for $YAlO_3$. (**c**) Single-pulse hysteresis loop of the ferroelectric memristor displaying clear voltage thresholds ($V_{th}^+$ and $V_{th}^-$). (**d**) Measurements of STDP in the ferroelectric memristor. Modulation of the device conductance ($\Delta G$) as a function of the delay ($\Delta t$) between pre- and post-synaptic spikes. Seven data sets were collected on the same device showing the reproducibility of the effect. The total length of each pre- and post-synaptic spike is 600 ns.

weakened. We emulate the spikes from pre- and post-neurons (sketched in Fig. 1a) by the waveforms shown in Fig. 1b: rectangular voltage pulses followed by smooth slopes of opposite polarity. Importantly, the voltage never exceeds $V_{th}$, so that a single spike cannot induce a change in resistance.

When both pre- and post-neuron spikes reach the memristor with a delay $\Delta t$, their superposition produces the waveforms $(V_{pre} - V_{post})$ displayed in the inset of Fig. 1d. The resulting combined waveform transitorily exceeds the threshold voltage, leading to an increase ($\Delta G > 0$, synapse strengthening) or a decrease ($\Delta G < 0$, synapse weakening) of the FTJ conductance ($G$), depending on the sign of $\Delta t$. As can be seen from the experimental STDP curve in Fig. 1d, only closely timed spikes produce a conductance change whereas long delays leave the device unchanged.

**Domain dynamics probed by tunnelling.** Modelling the shape of the STDP curve requires understanding the physical process underlying the time-dependent variation of the FTJ conductance while the waveform is applied. For this purpose, we image the pseudo real-time evolution of the ferroelectric domain configuration by means of stroboscopic piezoresponse force microscopy (PFM)[29], while simultaneously measuring the electrical properties of the device. Figure 2a shows the PFM phase and amplitude signals after cumulative pulses of constant amplitude (1 V) from the ON to the OFF state. The evolution of the phase images in stroboscopic PFM reveals the gradual reversal of the polarization from up (dark domains) to down (bright domains) and is reminiscent of the polarization reversal with increasing voltages previously observed by PFM[26]. The weak amplitude signal during reversal (for example, stages 2 and 3) indicates that the ferroelectric system is split into many small ($\lesssim 10$ nm) domains and that polarization switching is governed by the inhomogeneous nucleation of new domains rather than by the expansion of existing ones. We complemented these experiments by molecular dynamics (MD) simulations (Methods) on defect-free supertetragonal BFO films. These computations also yield an inhomogeneous character for polarization switching (Supplementary Fig. 1), therefore implying that this process is intrinsic in nature, that is, one does not require the presence of defects to obtain an inhomogeneous switching in supertetragonal BFO. Such findings contrast with the case of bulk-like BFO (rhombohedral phase) for which a homogeneous switching was recently predicted[30] as a result of large oxygen octahedra tilts that provide a specific homogeneous path for polarization reversal.

We extract the normalized reversed area $S$ from the phase images and plot it as a function of the cumulated pulse duration in Fig. 2b (black squares). Owing to the direct link between the junction resistance $R$ and this normalized reversed area $S$ (well described by a simple model of parallel resistances, $1/R = G = (1 - S) \times 1/R_{ON} + S \times 1/R_{OFF}$, ref. 25), one can also extract $S$ from measurements of the junction resistance after consecutive pulses of 1 V (green squares in Fig. 2b; note the excellent agreement between the PFM and transport data). Figure 2c shows transport data sets at different pulse amplitudes. They follow a systematic trend where, for a given cumulated pulse duration, the switched area is larger under higher voltages.

In ferroelectrics, inhomogeneous polarization switching can be described by a nucleation-limited model, which considers that the ferroelectric film is composed of different zones with independent switching kinetics[31,32]. Assuming for each voltage $V$ a broad Lorentzian distribution of the logarithm of nucleation times—with width $\Gamma(V)$ and centred at $\log(t_{mean}(V))$—the normalized reversed area $S$ can be approximated as a function of time $t$ and

applied voltage $V$ (ref. 31):

$$S_{\pm}(t, V) = \frac{1}{2} \mp \frac{1}{\pi} \arctan \frac{\log(t_{mean}(V)) - \log(t)}{\Gamma(V)}, \quad (1)$$

where the index relates to the sign of the applied voltage. Fits obtained with this expression accurately reproduce the experimental data of ferroelectric switching as a function of time and voltage (black lines in Fig. 2c). Figure 2d displays several representative distributions of switching times and illustrates the main trends. For larger voltages, switching occurs earlier and in a narrower time window (decrease of $t_{mean}$ and $\Gamma$), in agreement with previous results obtained on thick ferroelectric capacitors[32]. As shown in the inset of Fig. 2d, MD simulations performed at 10 K confirm the relevance of equation (1) (Methods and Supplementary Fig. 2) to characterize the switching process as well as the measured trends of $t_{mean}$ and $\Gamma$ with the magnitude of the electric field. Figure 2e shows that the evolution of the switching time ($t_{mean}$) as a function of the inverse electric field, extracted from transport measurements (Fig. 2c), follows the characteristic Merz's law[32] for ferroelectric switching ($t_{mean} \propto \exp(\alpha/E)$) where the activation field $\alpha$ is of the order of 3.0 V nm$^{-1}$. Due to the idealized nature of BFO during MD simulations (no interface, no defects, no tunnel current), the timescale for polarization switching is much shorter than in experiments, while the electric field is larger. Nevertheless, Merz's law can be applied to the MD simulation results (inset of Fig. 2e) and indicates an activation field of 2.4 V nm$^{-1}$, that is, in the same range as for experimental results. These simulation results strongly suggest that the experimentally observed inhomogeneous polarization switching in ultrathin films of BFO has an intrinsic origin.

**Predictive modelling of synaptic learning.** Because of the direct relationship between $S$ and $R$ in these devices, the accurate description of ferroelectric switching by the nucleation-limited model can be further extended to the modelling of resistance changes as a function of voltage amplitude. Figure 3a shows nested resistance hysteresis loops as a function of voltage pulse amplitude, characteristic of memristors. The STDP curve of the same device is displayed in Fig. 3b. Using equation (1) in combination with the parallel resistance model (Methods and Supplementary Fig. 3), we simultaneously fit the resistance versus voltage cycles and the STDP curve. Both resistance changes can be accurately replicated (solid lines in Fig. 3a,b) using the same set of parameters $t_{mean}(V)$ and $\Gamma(V)$.

This full description for each memristor device makes it possible to predict the conductance changes driving STDP learning in ferroelectric synapses. In Fig. 3b–d, we apply different voltage waveforms to our memristors to emulate various types of pre- and post-neuron activities[3]. This procedure allows the generation of biologically realistic[2], though accelerated (Fig. 3b,c), or artificially designed (Fig. 3d) STDP learning curves. Using our model with the parameters extracted previously, we can now predict the conductance changes for these specific types of STDP. Figure 3c,d shows the excellent agreement between these predictions and the measured conductance variations associated with different STDP waveforms.

## Discussion
We now use this physical model to simulate unsupervised learning in a spiking neural network with ferroelectric synapses. These simulations serve as a test bench to investigate the influence of the STDP waveform shape on the ability of the network to recognize patterns in images (here the horizontal, diagonal, and vertical bars labelled A, B and C, as shown in the

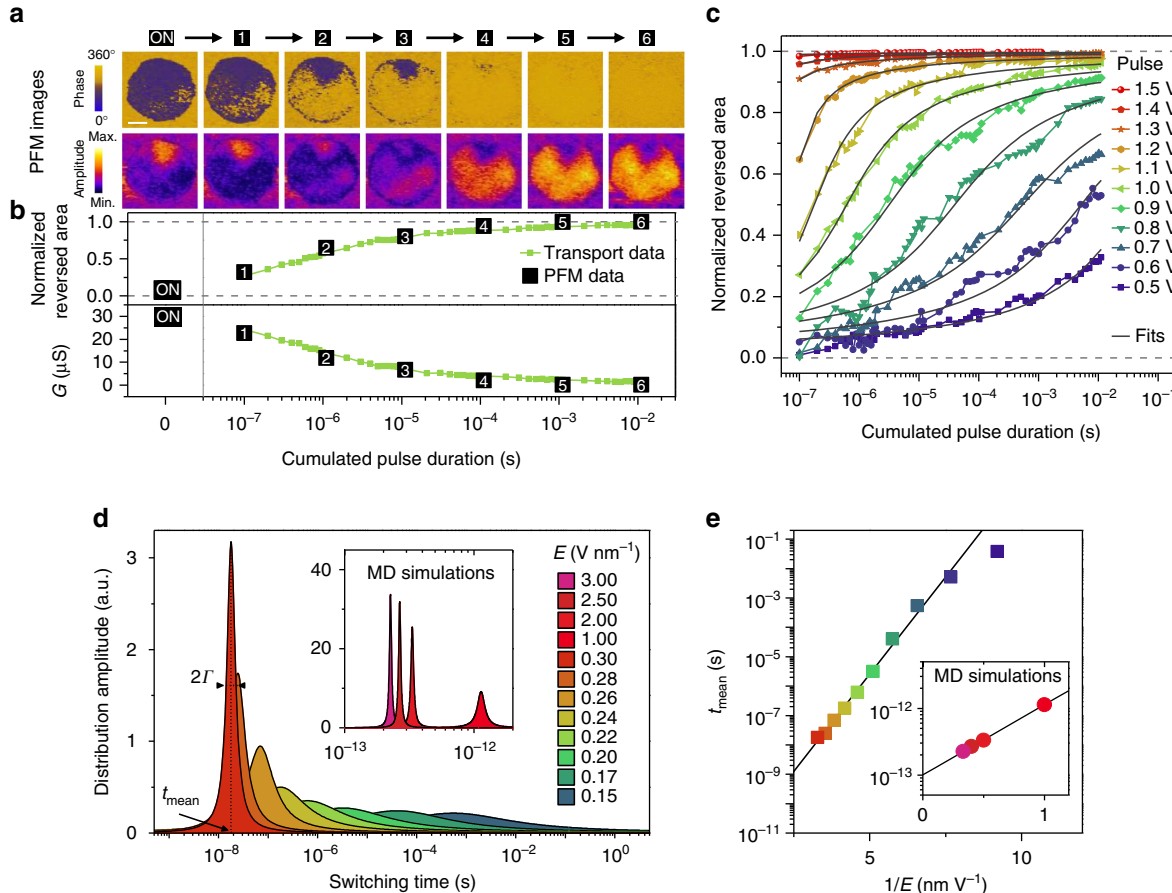

**Figure 2 | A memristor governed by nucleation-limited ferroelectric domain switching.** (**a**) Evolution of the PFM phase and amplitude signals of a ferroelectric memristor under cumulated pulses of 1 V and 100 ns. Cumulative pulses induce a progressive switching with multiple nucleation areas and limited propagation of ferroelectric domains from up (dark phase) to down (bright phase) polarization. The scale bar is 50 nm. (**b**, top) Normalized switched area as a function of cumulated pulse time calculated from time-dependent transport measurements of a ferroelectric memristor. The black squares indicate the normalized switched area obtained from the PFM measurements in **a**. (bottom) Corresponding conductance (G) evolution measured as a function of cumulated pulse time. (**c**) Normalized switched area as a function of cumulated pulse time calculated from time-dependent transport measurements of a ferroelectric memristor at different pulse amplitudes. The black lines are fit results from the nucleation-limited switching model. (**d**) Examples of Lorentzian distributions of switching times extracted from the fits in **c** at different pulse amplitudes and (inset) from the MD simulations (Supplementary Fig. 2). (**e**) Evolution of the switching time ($t_{mean}$) as a function of the inverse of the electric field (1/E) obtained from fits of the transport data in **c** and (inset) from MD simulations.

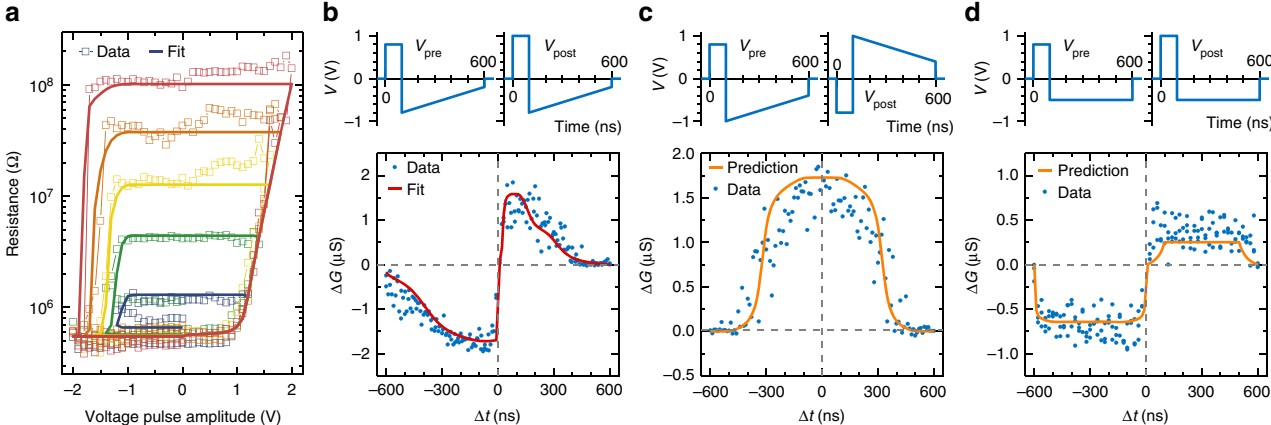

**Figure 3 | Predicting STDP learning with ferroelectric synapses.** (**a**) Multiple hysteresis loops of a ferroelectric memristor showing a clear dependence of resistance switching with the maximum pulse amplitudes. (**b**–**d**) Examples of STDP learning curves of different shapes. The pre- and post-synaptic spikes and conductance variations are shown in the top and bottom panels, respectively. For each device, simultaneous fits of data in **a**,**b** (solid lines) using equation (1) allow the prediction of new learning curves in **c**,**d** (orange lines).

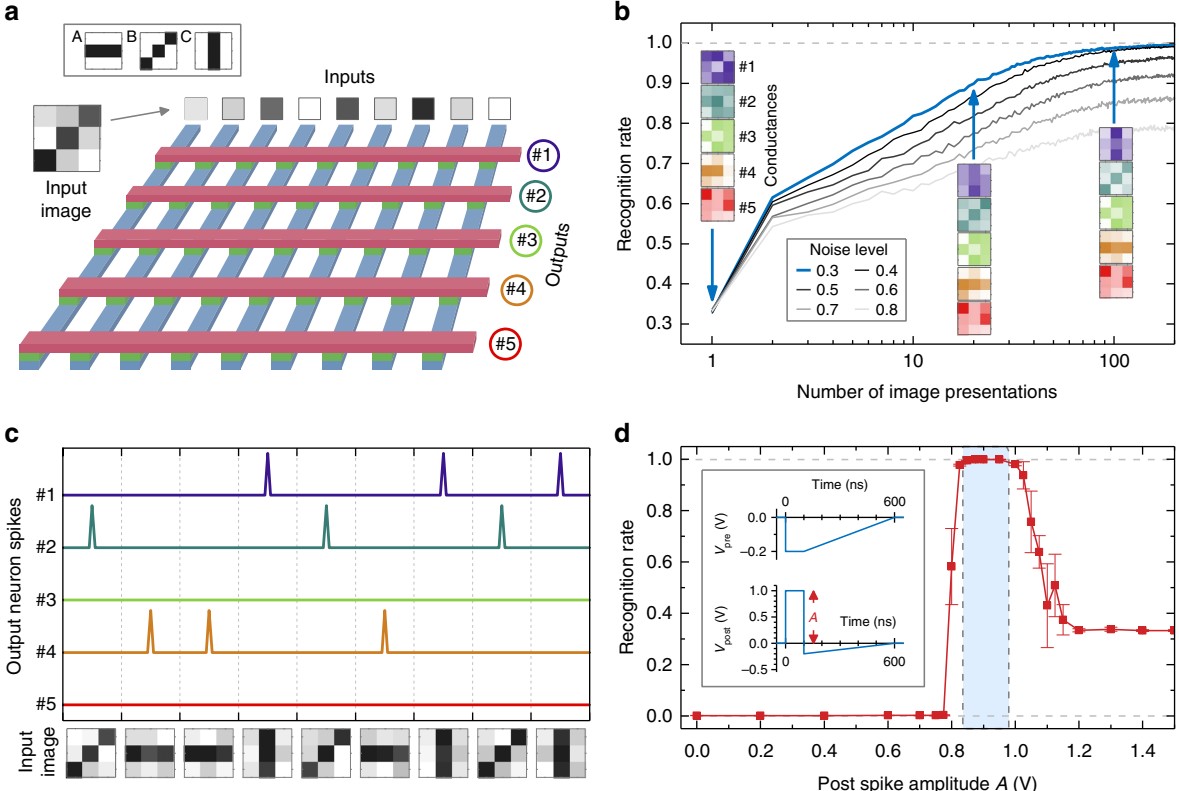

**Figure 4 | Unsupervised learning with ferroelectric synapses.** (**a**) Simulated spiking neural network comprising nine input neurons connected to five output neurons by an array of ferroelectric memristors. The inputs are noisy images of the patterns to recognize: horizontal (A), diagonal (B) and vertical (C) bars in $3 \times 3$ pixel images. (**b**) Recognition rate as a function of the number of presented images for different noise levels. The coloured images are conductance maps of the memristors in each line and show their evolution for a noise level of 0.3 (blue line). (**c**) Behaviour of the network after successful learning. Neurons 1, 2 and 4 emit spikes when inputs C, B and A are presented, respectively. (**d**) Evolution of the final recognition rate as a function of the amplitude of the post-synaptic spike. Error bars represent s.d.

inset of Fig. 4a). The simulated network is built around a crossbar of $9 \times 5$ ferroelectric memristors (Fig. 4a). Each of the nine spiking input neurons codes for one pixel of noisy images that contain one out of the three patterns to recognize (Methods). The output neurons integrate the input signals flowing through the memristors along each row and fire when a threshold is reached. Figure 4b shows that the recognition rate of the network increases with the number of image presentations, reaching 100% for low noise levels and almost 80% for high (of the order of the input amplitude) noise levels. The evolution of the conductance of the nine memristors in each row is shown in the inset of Fig. 4b as coloured $3 \times 3$ pixel images. As the network is learning, the conductance images in rows 1, 2 and 4 each converge toward one out of the three input patterns, whereas the memristor conductances in rows 3 and 5 remain random. After learning, as illustrated in Fig. 4c, three of the output neurons (neurons 1, 2 and 4 in this example) have specialized to each of the three input patterns and fire when the corresponding input is presented. Our simulations reveal that successful unsupervised learning is highly dependent on the exact shape of the pre- and post-neuron spike waveforms. Figure 4d shows that, for the case presented before, the recognition rate reaches 100% for well-chosen post-neuron spike amplitudes in the range of 0.82–0.98 V. However, when the amplitude of the post-neuron spike waveform is only slightly lower or higher, the recognition rate rapidly drops. Our simulations therefore emphasize the importance of a precise knowledge of the memristor dynamics, and therefore of its accurate description on the basis of a physical model.

In summary, we have established that STDP can be harnessed from intrinsically inhomogeneous polarization switching in ferroelectric memristors. Combining time-dependent transport measurements, ferroelectric domain imaging, and effective-Hamiltonian-based atomistic MD simulations, we show that the ferroelectric switching underlying resistive changes in these devices can be described by a well-established nucleation-limited model. Using this physical model, we can reliably predict the conductance evolution of ferroelectric synapses with varying neural inputs. These results pave the way toward low-power hardware implementations of billions of reliable and predictable artificial synapses[33] (such as deep neural networks[34]) in future brain-inspired computers.

## Methods

**Sample fabrication.** The BiFeO₃ (4.6 nm)/Ca₀.₉₆Ce₀.₀₄MnO₃ (20 nm) thin films are grown on YAlO₃ (YAO) (001) substrates by pulsed laser deposition using a Nd:YAG laser (details of the growth can be found in ref. 26). The thin films are epitaxial and fully strained by the substrates as characterized by X-ray diffraction experiments[26] and scanning transmission electron microscopy on cross-section samples[35]. The high compressive strain imposed by the YAO substrate on BiFeO₃ (BFO) stabilizes the supertetragonal polymorph of BFO with giant tetragonality and a potentially large polarization pointing initially towards the bottom electrode of Ca₀.₉₆Ce₀.₀₄MnO₃ (CCMO)[35].

FTJs are then fabricated by defining Pt $(10 - 90\,\text{nm})$/Co $(5 - 10\,\text{nm})$ nanopillar electrodes (with diameters of 180–500 nm) on top of the unpatterned BFO/CCMO heterostructure, combining conventional electron-beam lithography and lift-off processes. For experiments involving electron transport and ferroelectric domain imaging experiments (as in Fig. 2), arrays of nanopillars are electrically contacted with a conducting atomic force microscopy tip[26]. For all other electronic transport experiments presented in the manuscript, fully patterned junctions are defined by a four-step lithography process following the definition of these nanopillars (details

of the fabrication process can be found in ref. 27). Such fully patterned junctions are macroscopically connected by standard radio frequency (RF) probes.

**Physical measurements.** Combined transport and PFM measurements are performed using solid-state FTJs with diameters of 180 nm, as detailed in ref. 26. These Pt/Co/BFO/CCMO junctions are connected electrically by a conductive atomic force microscopy tip to allow resistance measurements under constant dc voltage ($-0.1$ V, time constant 100 ms) and the application of 100-ns write pulses in combination with subsequent PFM imaging ($V_{ac} = 0.6$ V, $V_{dc} = -0.17$ V, $f = 14$ kHz). The bias voltage is applied to the tip while the bottom electrode is grounded.

The resistance–voltage cycles and STDP curves were obtained on fully processed FTJs with diameters of 400 and 500 nm, described in ref. 27. The resistance of the junctions was measured after each pulse at a low bias of $-200$ mV (time constant 100 ms). To obtain the STDP curve, we always initialize the junction to the ON state for $\Delta t < 0$ and to an intermediate resistance state between ON and OFF for $\Delta t > 0$. The exact values of initial resistance states in each figure are

Figs 1d and 3b: $R_{init}(\Delta t < 0) = 6 \times 10^5\ \Omega$, $R_{init}(\Delta t > 0) = 1.2 \times 10^7\ \Omega$,
Fig. 3c: $R_{init} = 1.2 \times 10^7\ \Omega$,
Fig. 3d: $R_{init}(\Delta t < 0) = 6 \times 10^5\ \Omega$, $R_{init}(\Delta t > 0) = 1.2 \times 10^7\ \Omega$.

**Molecular dynamics simulations.** The switching process under dc electric fields is simulated using a recently developed scheme[36], in which the total energy is provided by the first-principles-based effective Hamiltonian method of refs 37,38 and inserted into an original MD technique treating on the same footing dynamics of ions and magnetic degrees of freedom. A periodic $48 \times 48 \times 6$ supercell (69,120 atoms) of the supertetragonal BFO (P4mm space group) is adopted and the electric field is constantly applied along the pseudocubic [00$\bar{1}$] direction (opposite to the initial polarization direction). The temperature is kept at 10 K so as to minimize thermal fluctuations and the time step is set to 0.5 fs. Supplementary Fig. 1a shows polarization patterns at different times under a dc electric field of $2.5$ V nm$^{-1}$, which illustrate the (intrinsic) inhomogeneous switching of polarization in bulk-like supertetragonal BFO. The switched area as function of time is plotted in Supplementary Fig. 1b. Note that the reversed area is defined as any polarization that has a magnitude larger than $7\ \mu$C cm$^{-2}$ (it reaches $100\ \mu$C cm$^{-2}$ when being fully switched) in the [00$\bar{1}$] direction. Simulation results of polarization switching under different electric fields are summarized in Supplementary Fig. 2. Fits from equation (1) are represented as black lines. Lorentzian switching distributions resulting from the fits are displayed in the inset of Fig. 2d and the switching times are plotted in the inset of Fig. 2e.

**Domain dynamics model.** The nucleation-limited model as expressed by equation (1) describes the full reversal of the ferroelectric domains starting from a uniform state under constant voltage amplitude. To describe the ferroelectric switching dynamics under the influence of arbitrary voltage waveforms, the waveform is numerically divided in short segments of width $\Delta t$ (Supplementary Fig. 3a). For sufficiently small $\Delta t$, we can then assume a constant applied voltage $V_i$. As the reversal dynamics depends on the present domain configuration and therefore on the value $S_{i-1}$ before the segment $i$, we compute the time offset $t_i^{offset}$ corresponding to a voltage $V_i$ and can only then apply equation (1) to calculate the value $S_i = S(t_i^{offset} + \Delta t, V_i)$ (Supplementary Fig. 3b). Owing to the highly non-linear reversal dynamics of FTJs where high voltage amplitudes govern the reversal process, the value of $\Delta t$ has to be chosen small enough not to smear out these maximal amplitudes. For the results presented here, we chose $\Delta t \leq 20$ ns.

A simple yet rigorous model connects the electrical resistance $R$ with the ferroelectric domain configuration characterized by the normalized area of down domains $S$. Assuming parallel conduction through areas with ferroelectric down domains and areas with up domains allows to calculate one from another using the relation $1/R = (1 - S) \times 1/R_{ON} + S \times 1/R_{OFF}$, where $R_{ON}$ and $R_{OFF}$ are the resistance values in the low- and high-resistance states, respectively.

**Spiking neural network simulations.** The simulated spiking neural network consists of nine input neurons fully connected to five output neurons by a crossbar of $9 \times 5$ identical ferroelectric memristors, as depicted in Fig. 4a. Inputs are consecutively presented to the network in the form of grey-level images of $3 \times 3$ pixels composed of one of the three patterns to recognize (horizontal, diagonal, and vertical bars labelled A, B and C in the inset of Fig. 4a) and an additional uniformly distributed random noise of the specified amplitude. Subsequently, the values of each image are normalized to the interval [0, 1]. Each individual pixel corresponds to one input neuron that integrates the respective input value over time (using a fixed time step) and spikes once it reaches a certain threshold $Q_{th}$. This implementation effectively results in a frequency coding of the input pixel values. The spikes from each input neuron are transmitted to all output neurons (as in a fully connected network) that integrate them weighted by the corresponding connection strength, that is, by the normalized conductance value of the respective FTJ. Once an output neuron (#1 to #5 in Fig. 4) reaches the threshold value $Q_{th}$, it spikes on its part and generates an output spike. Lateral inhibition then resets all output neurons. We use the first output spike after the presentation of a new input image to determine if the network has successfully recognized the pattern (A, B or C as

shown in Fig. 4a) of the input image. As for unsupervised learning the labels of the output neurons are not known beforehand, the labelling is done algorithmically for each repetition of the simulation. We define a successful learning as when each pattern is immediately recognized after 200 input image presentations. The recognition rates shown in Fig. 4b,d are calculated as the mean of 100 simulation runs for each parameter set. At the beginning of each simulation run, the memristor conductances are initialized to random values between those of their ON and OFF states. Variations between the runs using identical parameters are therefore due to the randomized initial conductance values of the synapses as well as the random noise that is added to the input patterns.

**Data availability.** The data that support the findings of this study are available from the corresponding authors upon request.

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

## Acknowledgements

We thank Cyrile Deranlot and Eric Jacquet for assistance with the sample fabrication and D.J. Kim for fruitful discussions. Financial support from the European Research Council (ERC Advanced Grant No. 267579) and the French Agence Nationale de la Recherche (ANR) through projects FERROMON and MIRA are acknowledged. This publication has received funding from the European Union's Horizon 2020 research innovation programme under grant agreement 732642 (ULPEC project). This work is supported by a public grant overseen by the French National Research Agency (ANR) as part of the 'Investissements d'Avenir' program (Labex NanoSaclay, reference: ANR-10-LABX-0035). B.X. and L.B. acknowledge the financial support from the Department of Energy, Office of Basic Energy Sciences, under contract ER-46612, and DARPA grant HR0011-15-2-0038 (MATRIX program), respectively.

## Author contributions

J.G., S.S. and V.G. designed and organized the experiments and calculations. S.G., C.C., K.G. and S.X. elaborated the thin films and nanofabricated the samples. S.F. and V.G. performed the piezoresponse force microscopy experiments. S.B., J.G., G.L., S.F., J.T., S.S. and V.G. performed the electrical transport measurements. B.X. performed the atomic-scale molecular dynamics simulations, which were then discussed with L.B. S.B., N.L. and J.G. performed the spiking neural network simulations. All the authors contributed to the interpretation of the results. S.B., J.G. and V.G. wrote the manuscript with input from all authors.

## Additional information

**Competing interests:** The authors declare no competing financial interests.

**Publisher's note**: 

