## [Peer Review File · Nature Communications]

Reviewers' comments:

Reviewer #1 (Remarks to the Author):

The authors demonstrate synapses based on the ferroelectric tunnel junctions for autonomous learning. This work could be further extended for the future implementation of the unsupervised learning because well-established physical principles were used to explain operational mechanisms. This is a very interesting work. I would recommend its publication if following issues can be clarified. In particular, there should be more explanation for some parts of the manuscript.

1. In line 79-80, there is no explanation for synapse strengthening ΔG . It is probably the change in the memristor conductance.

2. Figure 2a and b show the direct link between the junction resistance from transport data and the normalized reserved area from PFM. Because this direct link was shown in Figure 2b, it was possible to apply well-established physical principles. Typically, an application of dc voltage is not necessary for PFM imaging. Why did the authors apply dc voltage for PFM imaging?

3. The experimental results for 0.5 and 0.6 V in Figure 2e do not follow well the fit. In fact, the authors applied relatively high ac voltage, 0.6 V. It may be hard to get reliable data for at least 0.5 and 0.6 V.

4. The MD simulations were performed at 10 K, however the experiments might be performed at RT. At different temperatures, the switching time can be different (please see Figure 1 in ref. 32). Did the authors consider this point in Figure 2e? In fact, because the authors already explained it using a well-established physical principle of eq. 1, it may not be necessary to add the MD simulations in the manuscript.

5. In Methods, there is only conceptual description for Figure 4. There should be more explanation. How did the authors actually perform spiking neural network simulations? What is the unit of y axis in Figure 4c? What are input and output signals? How can the authors calculate recognition rate?

Reviewer #2 (Remarks to the Author):

This manuscript reports the exploitation of Ferroelectric Tunnel Junction (FTJ) memristors to mimic synapse behaviour, by tailoring pre and post electrical pulses to trigger different degrees of ferroelectric switching depending on the delay between their application. The resistance of the FTJ will change only if the pulses are sufficiently close in time and will either induce progressively lower resistance (stronger synapse strengths) if the post pulse occurs shortly after the pre-pulse (causality), or higher memristor resistance (weaker synapse strength) if the pre-pulse occurs shortly after the post-pulse (anticausality). The authors go on to demonstrate how a simple self-learning image recognition system might be constructed.

The manuscript is extremely exciting and fully merits publication in Nature Comms. There are a few issues that I would raise, not because there are any flaws in the science performed, but because there were points in the story where I became a little confused and perhaps greater clarity of communication could be developed (either through slight alterations in the manuscript itself, or through more extensive supplementary information):

(i) it took me a while to make the connection absolutely clear between low resistance in the

memristor and stronger synapse strength (and equally, high resistance and weak synapse strength); the connection is there, but perhaps it could be made even more explicit for the readership that isn't so quick on the uptake !

(ii) I was confused by the schematic pulses in figure 1a - it wasn't clear to me that there was a difference in time between the two pulses sketched - perhaps this could be altered a little to be more instructive - perhaps an explicit example of causality in pulses delivered to a real synapse;

(iii) the net voltage functions given in figure 1d are a little confusing and here I would recommend an explicit section in the supplementary information: for example, if I take the pulses given in figure 1b and add them together with a very small Δt (smaller than the width of the initial square-pulse section of the individual pulse function), then an initial square section of height V_{+max} will increase to another square section of height $2V_{+max}$, before collapsing to close to zero and then becoming negative at close to $2V_{-max}$ and ramping back to zero.....it is not at all clear that the small square section at $2V_{+max}$ should be lower than the almost $2V_{-max}$ section and therefore it looks to be that a region of behaviour exists where the memristor first switches on the positive side and then later switches past the negative threshold. I agree that, in the end, the final state would be a negative switch state, but it is obvious that the situation is potentially a lot more complex than the sketched resultant waveform presented in the figure at the moment. I would recommend some explicit function additions for a range of Δt to back up this initial figure (and put this in the supplementary section).

(iv) it is clear that the net synapse behaviour is completely dictated by the choice of function used for pre and post synapse excitation. At the moment, it seems that the design of the pulses was done by intuition, but it would be great if the authors could offer a more formal and explicit mathematical transformation that could allow any synapse response function to be represented by its pre / post pulse function. This could be tricky, but it would be really useful and interesting.

Reviewer #3 (Remarks to the Author):

The authors present a ferroelectric based memristor. Normally, ferroelectric materials settle to a state dependent on the amplitude of an applied voltage pulse or signal. However, for neuro-inspired learning rules, it is necessary to have incremental increases for the same pulse amplitude. The device presented by the authors has this property, if the width of the applied pulses is very short, in the range of hundreds of nanoseconds (or even a few microseconds).

The authors present in-depth device characterizations, explanations about their internal physics with precise modeling capable of predicting the measured behavior. The authors then use these model to simulate a neural system capable of learning and retrieving patterns. To my knowledge, this has not been reported before as such, and the work is of very high quality, therefore deserving publication. However, I have some comments to be considered before final publication, in order to clarify some issues within the perspective of future application in realistic neuromorphic CMOS-memristor systems:

1.- I am not expert in device physics, so this question is pure curiosity: is the ferroelectric tunneling junction physics subject to aging? Could you comment on this in the paper?

2.- Page 6, line 129, "... relationship between S and R in ...". Shouldn't it be "... relationship between S and G in ..."?

3.- Fig.2b. You show normalized reverse area versus cumulated pulse duration. You should also add measured R or G.

4.- Figs.3b-d. Please explain how you obtained these ΔG measurements? Did you reset every time to an initial G_0 ? Or is it cumulative? How does ΔG depend on the instantaneous G? Note

that these are key issues for STDP. Can you state if you have additive STDP, multiplicative STDP, exponential (fractional) STDP, or combined?

5.- Page 7, line 140, "This procedure allows the generation of biologically realistic (Fig. 3b and c) or artificially designed (Fig. 3d) STDP learning curves. Well, the obtained STDP is biologically realistic, except for the time scales involved. Here, the STDP overlap time window is in the order of +/-600ns. In biological synapses, instead of 600ns you would have 60ms to 100ms: 5 orders of magnitude off. This means that your devices could be useful for accelerated neuromorphic systems (assuming the rest of components, besides the synapses, could be made to operate at such acceleration). But, for example, to use them in a robotic self-learning system that operates and interacts with the environment in real time, they could not be used, at least in the way STDP is described in the paper. You should clarify this as well in the paper.

6.- In page 8, lines 162-165 you mention that recognition performance is critically dependent on spike amplitudes. This highlights the fact that recognition might be also strongly dependent on other parameters as well, and mostly on inter-device parametric mismatch. Since you have a very precise model for the device operation, and have fabricated several devices, there should be a statistical characterization on the variations of the different parameters from device to device. Since the physics is based on ferroelectric phenomena, I would expect (in principle) that the amplitude of spikes is not a critical issue. However, pulse nano-second widths for incremental changes might impact on the mismatch between different devices (this is, two devices may need very different pulse widths to be subject to the same increment/decrement in conductance G). Additionally, there might be other parameters within the devices subject to high inter-device mismatch. Consequently, statistical mismatch characterization is crucial, and I believe the authors have sufficient data to at least provide a Table on this.

7.- Having high R_{on} (300Kohm) and R_{off} (100Mohm) are a plus for low power. However, in your case, you need also fast dynamics. So, care must be taken with parasitic capacitances. For example, suppose you want to implement a large array. Your STDP dynamics are in the order hundreds of ns. So, you need your currents and spikes to be collected by circuits with parasitic time constants below this, say 10ns. Therefore, the RC time constants of your array lines must be $RC=10ns$ (or less). Let's consider as worst case when one resistance is at min value, about 300Kohm from Fig.3a, and the rest at max 100Mohm. Then $RC<10ns$ implies that $C<30fF$ (neglecting the impact of the 100Mohm resistances). In present day CMOS technologies, long lines tend to have parasitic capacitances in the range of pF. Of course, you plan to have very dense arrays of nano scale synapses, but there will be a limit on scalability determined by your fast STDP time constants, max and min resistances, and line parasitic capacitances. You should include a Section to discuss about what this limit would be.

Reviewers' comments:

Reviewer #1 (Remarks to the Author):

#0. The authors demonstrate synapses based on the ferroelectric tunnel junctions for autonomous learning. This work could be further extended for the future implementation of the unsupervised learning because well-established physical principles were used to explain operational mechanisms. This is a very interesting work. I would recommend its publication if following issues can be clarified. In particular, there should be more explanation for some parts of the manuscript.

We thank Reviewer #1 for his/her positive comments. In the following we address point by point the issues he/she raised.

#1. In line 79-80, there is no explanation for synapse strengthening ΔG . It is probably the change in the memristor conductance.

We apologize if this sentence was not clear enough. We rephrased it to: "The resulting combined waveform transitorily exceeds the threshold voltage, leading to an increase ($\Delta G > 0$, synapse strengthening) or a decrease ($\Delta G < 0$, synapse weakening) of the FTJ conductance (G), depending on the sign of Δt ."

#2. Figure 2a and b show the direct link between the junction resistance from transport data and the normalized reserved area from PFM. Because this direct link was shown in Figure 2b, it was possible to apply well-established physical principles. Typically, an application of dc voltage is not necessary for PFM imaging. Why did the authors apply dc voltage for PFM imaging?

The Reviewer is right when mentioning that in principle a dc voltage is not necessary for PFM imaging. However, the application of a significant ac voltage, necessary to read the PFM response while imaging, can sometimes destabilize the ferroelectric configuration and corresponding tunnel resistance in the case of ferroelectric devices with small imprint (small horizontal shift in Figure 1c). Therefore we applied a small dc voltage to compensate this imprint [see Gruverman *et al.*, Appl. Phys. Lett. 87, 082902 (2005)]. We found that applying a small dc voltage of -170 mV to the top electrode together with the ac voltage of 600 mV for PFM imaging leads to virtually no resistance change: we always measured the tunnel resistance of the junctions before and after PFM imaging to check that the ferroelectric state of the junction was not disturbed by PFM imaging.

#3. The experimental results for 0.5 and 0.6 V in Figure 2e do not follow well the fit. In fact, the authors applied relatively high ac voltage, 0.6 V. It may be hard to get reliable data for at least 0.5 and 0.6 V.

We agree with the Reviewer that the linear fit in Figure 2e is not intercepting the data for voltage pulse amplitudes of 0.5 and 0.6 V. However, we do not understand his/her point regarding the possible role of the PFM ac amplitude of 0.6 V in these measurements as PFM measurements were not involved here. We would like to emphasize that the normalised reversed area (S) evolution with time (Figure 2c) is determined only from the tunnel resistance evolution measured under cumulative pulses using the parallel resistance formula $1/R = (1 - S) \times 1/R_{ON} + S \times 1/R_{OFF}$. We fitted this time evolution of the

normalised reversed area with the nucleation-limited model displayed in Eq. (1) and extracted a Lorentzian distribution of the switching times for each voltage pulse amplitude (Figure 2d). For low voltage pulses of 0.5 and 0.6 V in Figure 2c, the polarisation switching is incomplete even after 10^{-2} s of cumulated pulses (i.e., 10^5 pulses of 100 ns each) and yields a very broad distribution of waiting times (see Figure 2d). In these specific cases where the variation of S is small, the fits from Eq. (1) contain much more uncertainties that could explain the discrepancy of the mean switching times in Figure 2e.

#4. The MD simulations were performed at 10 K, however the experiments might be performed at RT. At different temperatures, the switching time can be different (please see Figure 1 in ref. 32). Did the authors consider this point in Figure 2e? In fact, because the authors already explained it using a well-established physical principle of eq. 1, it may not be necessary to add the MD simulations in the manuscript.

We agree with the Reviewer that temperature can affect the switching time. For instance, additional MD simulations we performed at 300 K indeed yield shorter switching times. However, the switching mechanism is essentially identical to that at 10 K, i.e., inhomogeneous switching described by the nucleation-limited model of Eq. (1). Note that we practically choose 10 K in the MD simulations to reduce the amount of dipolar flipping due to thermal effects alone and thus to investigate dipolar flipping mostly related to the sole application of electric fields. Please also note that the MD results are provided in this manuscript to a less extent to emphasize the quantitative comparison with the switching time from experiment (even if the characteristic times we computationally obtained agree well with the interpolated ones from experiments in Figure 2e), but more importantly to show that the inhomogeneous switching governed by Eq. (1) can intrinsically occur in the T-phase BFO. To emphasize this point, we modified the sentence line 214: " Supplementary Fig. 1a shows polarisation patterns at different times under a dc electric field of 2.5 V nm^{-1} which illustrate the (intrinsic) inhomogeneous switching of polarisation in bulk-like supertetragonal BFO."

#5. In Methods, there is only conceptual description for Figure 4. There should be more explanation. How did the authors actually perform spiking neural network simulations? What is the unit of y axis in Figure 4c? What are input and output signals? How can the authors calculate recognition rate?

Figure 4a illustrates the simulated neural network. The inputs of the network are 3×3 pixels greyscale images. Images are presented one at a time, successively. There is a one to one correspondence between the 9 pixels of the input image, and the 9 input neurons. Each pixel is attributed a value between 0 and 1 according to its colour. The corresponding neuron integrates this input value over time (using a fixed time step) and spikes once it reaches a certain threshold Q_{th} . Therefore, during the presentation time of the image the 9 input neurons emit voltage spikes to the bottom electrodes (blue columns in Fig. 4a) with a frequency proportional to the intensity of the input pixel they code for. The spikes from each input neuron are transmitted to all output neurons.

The current flowing to output neuron #i is the sum of the voltage spikes incoming from input neurons weighted by the conductance values of ferroelectric tunnel junctions in row i (rows are in pink in Fig. 4a). The output neurons integrate this signal, and emit a spike once they reach the threshold value Q_{th} . Lateral inhibition then resets all output neurons.

Initially, the memristor conductances in the array are set to random values between the ON and OFF states. Then, they evolve according to the STDP mechanism. More precisely, the value of a memristor located in the crossbar at the intersection of column i and row $\#j$ evolves according to the relative timing between the spikes emitted by the input neuron i and the output neuron $\#j$ (the waveforms corresponding to pre and post spikes are shown in the inset of Fig. 4d).

We use the first output spike after the presentation of a new input image to determine if the network has successfully recognised the pattern (A, B, or C as shown in Figure 4a) of the input image. For example, if neuron $\#i$ spikes first, we compute the distances between the normalized weights in row $\#i$ and the pixels values of patterns A, B and C. If the closest pattern corresponds to the pattern in the noisy input image, the network has successfully recognized the image. Otherwise, neuron $\#i$ is spiking for wrong reasons: the recognition has failed. We define a successful learning as when each pattern is immediately recognized after 200 input image presentations.

In Figure 4c we schematically visualize the functioning of the simulated neural network after successful learning. The y axis of Figure 4c uses an arbitrary unit and only indicates the presence (or absence, respectively) of a spike of each of the output neurons individually. We changed the description scheme of the output neurons in Figure 4 to clarify this.

The recognition rates shown in Figure 4b and d are calculated as the mean of 100 simulation runs for each parameter set. Variations between the runs using identical parameters are due to the randomised initial conductance values of the synapses as well as the random noise that is added to the input patterns.

In order to give more explanation on how the simulations were performed we modified the Methods section on “Spiking neural network simulations” to:

“The simulated spiking neural network consists of 9 input neurons fully connected to 5 output neurons by a crossbar of 9×5 identical ferroelectric memristors, as depicted in Fig. 4a. Inputs are consecutively presented to the network in the form of grey-level images of 3×3 pixels composed of one of the three patterns to recognise (horizontal, diagonal, and vertical bars labelled A, B, and C in the inset of Fig. 4a) and additional uniformly distributed random noise of the specified amplitude. Subsequently, the values of each image are normalised to the interval $[0, 1]$. Each individual pixel corresponds to one input neuron that integrates the respective input value over time (using a fixed time step) and spikes once it reaches a certain threshold Q_{th} . This implementation effectively results in a frequency coding of the input pixel values. The spikes from each input neuron are transmitted to all output neurons (as in a fully connected network) that integrate them weighted by the corresponding connection strength, i.e., by the normalised conductance value of the respective ferroelectric tunnel junction. Once an output neuron ($\#1$ to $\#5$ in Fig. 4) reaches the threshold value Q_{th} , it spikes on its part and generates an output spike. Lateral inhibition then resets all output neurons. We use the first output spike after the presentation of a new input image to determine if the network has successfully recognised the pattern (A, B, or C as shown in Figure 4a) of the input image. As for unsupervised learning the labels of the output neurons are not known beforehand, the labelling is done algorithmically for each repetition of the simulation. We define

a successful learning as when each pattern is immediately recognized after 200 input image presentations. The recognition rates shown in Fig. 4b and 4d are calculated as the mean of 100 simulation runs for each parameter set. At the beginning of each simulation run, the memristor conductances are initialised to random values between those of their ON and OFF states. Variations between the runs using identical parameters are therefore due to the randomised initial conductance values of the synapses as well as the random noise that is added to the input patterns.”

Reviewer #2 (Remarks to the Author):

#0. This manuscript reports the exploitation of Ferroelectric Tunnel Junction (FTJ) memristors to mimic synapse behaviour, by tailoring pre and post electrical pulses to trigger different degrees of ferroelectric switching depending on the delay between their application. The resistance of the FTJ will change only if the pulses are sufficiently close in time and will either induce progressively lower resistance (stronger synapse strengths) if the post pulse occurs shortly after the pre-pulse (causality), or higher memristor resistance (weaker synapse strength) if the pre-pulse occurs shortly after the post-pulse (anticausality). The authors go on to demonstrate how a simple self-learning image recognition system might be constructed.

The manuscript is extremely exciting and fully merits publication in Nature Comms. There are a few issues that I would raise, not because there are any flaws in the science performed, but because there were points in the story where I became a little confused and perhaps greater clarity of communication could be developed (either through slight alterations in the manuscript itself, or through more extensive supplementary information):

We thank the Reviewer for his/her enthusiasm regarding the exciting character of this manuscript. In the following, we will clarify the points he/she addressed.

#1. It took me a while to make the connection absolutely clear between low resistance in the memristor and stronger synapse strength (and equally, high resistance and weak synapse strength); the connection is there, but perhaps it could be made even more explicit for the readership that isn't so quick on the uptake!

We understand the Reviewer's concerns about the lack of explicit relationship between the synaptic strength and memristor conductance in the introduction of the manuscript. In the revised version of the manuscript, we modified the sentence line 46: “Memristors thus exhibit plasticity and their conductance can emulate synaptic strength, so that a low resistance corresponds to a strong synaptic connection and a high resistance corresponds to a weak synaptic connection, respectively.”

#2. I was confused by the schematic pulses in figure 1a - it wasn't clear to me that there was a difference in time between the two pulses sketched - perhaps this could be altered a little to be more instructive - perhaps an explicit example of causality in pulses delivered to a real synapse;

The Reviewer is right when suggesting the neuron spikes in Figure 1a could be shifted in time. Actually there was not initially any time difference between the post- and pre-synaptic neuron spikes in the Figure. We have modified the Figure and its caption according to the Reviewer's recommendations.

#3. The net voltage functions given in figure 1d are a little confusing and here I would recommend an explicit section in the supplementary information: for example, if I take the pulses given in figure 1b and add them together with a very small delta t (smaller than the width of the initial square-pulse section of the individual pulse function), then an initial square section of height V_{+max} will increase to another square section of height $2V_{+max}$, before collapsing to close to zero and then becoming negative at close to $2V_{-max}$ and ramping back to zero.....it is not at all clear that the small square section at $2V_{+max}$ should be lower than the almost $2V_{-max}$ section and therefore it looks to be that a region of behaviour exists where the memristor first switches on the positive side and then later switches past the negative threshold. I agree that, in the end, the final state would be a negative switch state, but it is obvious that the situation is potentially a lot more complex than the sketched resultant waveform presented in the figure at the moment. I would recommend some explicit function additions for a range of delta t to back up this initial figure (and put this in the supplementary section).

We apologize that the definition of the voltage waveform was not clear enough in the previous version of the manuscript. The Reviewer has to take into account that the pre- and post-synaptic voltage signals in Figure 1b are applied to the top and bottom electrode, respectively. Therefore the resulting waveform in Figure 1d is the difference between the two signals: ($V_{pre} - V_{post}$). Then, the square sections of the two inputs never add up to reach twice the maximum voltage signal but tend to cancel each other at zero time difference. For example, in Figure 1d, for small positive time delays, the negative square waveform of the post-synaptic spike adds up to the pre-synaptic negative tail to reach the negative voltage threshold of the ferroelectric memristor. We modified the sentence line 76: "When both pre- and post-neuron spikes reach the memristor with a delay Δt , their superposition ($V_{pre} - V_{post}$) produces the waveforms displayed in the inset of Fig. 1d"

#4. It is clear that the net synapse behaviour is completely dictated by the choice of function used for pre and post synapse excitation. At the moment, it seems that the design of the pulses was done by intuition, but it would be great if the authors could offer a more formal and explicit mathematical transformation that could allow any synapse response function to be represented by its pre / post pulse function. This could be tricky, but it would be really useful and interesting.

We agree with the Reviewer that a simple, explicit formula to predict the synaptic response, i.e., the FTJ conductance changes, to a given spike shape would indeed be convenient. However, the dynamics of polarization reversal can only be expressed in a closed form for constant bias, as given in Eq. 1. In order to perform large-scale simulations of neural networks based on FTJs, one would eventually try to represent the FTJ conductance change response by an explicit formula. However, this would also come at the expense of losing the physics of polarization reversal behind it. We refrain from doing this here as it would not fit the scope of this paper.

Reviewer #3 (Remarks to the Author):

#0. The authors present a ferroelectric based memristor. Normally, ferroelectric materials settle to a state dependent on the amplitude of an applied voltage pulse or signal. However, for neuro-inspired learning rules, it is necessary to have incremental increases for the same pulse amplitude. The device presented by the authors has this property, if the width of the applied pulses is very short, in the range of hundreds of nanoseconds (or even a few microseconds).

The authors present in-depth device characterizations, explanations about their internal physics with precise modeling capable of predicting the measured behavior. The authors then use these model to simulate a neural system capable of learning and retrieving patterns. To my knowledge, this has not been reported before as such, and the work is of very high quality, therefore deserving publication. However, I have some comments to be considered before final publication, in order to clarify some issues within the perspective of future application in realistic neuromorphic CMOS-memristor systems:

We are grateful to the Reviewer for his/her positive comments regarding our manuscript. In the following we answer point by point the issues he/she raised.

#1. I am not expert in device physics, so this question is pure curiosity: is the ferroelectric tunneling junction physics subject to aging? Could you comment on this in the paper?

If by aging, the Reviewer refers to fatigue related to multiple cycling of polarisation, we would like to mention that we demonstrated good endurance properties with more than 4×10^6 cycles [Boyn et al. Applied Physics Letters 104, 52909 (2014)] using the same ferroelectric tunnel junctions as those described in this manuscript. This issue is addressed in the sentence line 62: "Furthermore, supertetragonal BiFeO₃ (BFO) tunnel barriers combined with (Ca,Ce)MnO₃ (CCMO) bottom and Co top electrodes give rise to OFF/ON resistance ratios up to 10^4 (ref. 26) paired with high endurance and operation speed [27]."

#2. Page 6, line 129, "... relationship between S and R in ...". Shouldn't it be "... relationship between S and G in ..."?

We think the initial sentence is accurate as the direct relationship between S and R is described by parallel resistance formula. In order to avoid any confusion, we modified the sentence line 101: "Owing to the direct link between the junction resistance R and this normalised reversed area S (well described by a simple model of parallel resistances, $1/R = G = (1 - S) \times 1/R_{ON} + S \times 1/R_{OFF}$ ref. ²⁵), one can also extract S from measurements of the junction resistance..."

#3. Fig.2b. You show normalized reverse area versus cumulated pulse duration. You should also add measured R or G.

As suggested by the Reviewer, we added the measured conductance vs. cumulated pulses corresponding to the normalised reversed area vs. time in the revised version of Figure 2b. We modified the caption accordingly.

#4. Figs.3b-d. Please explain how you obtained these Delta_G measurements? Did you reset every time to an initial Go? Or is it cumulative? How does Delta_G depend on the instantaneous G? Note that these are key issues for STDP. Can you state if you have additive STDP, multiplicative STDP, exponential (fractional) STDP, or combined?

In the experiments presented in the manuscript, the conductance variations of the ferroelectric memristors are measured after resetting the resistance to specific values for positive and negative delay times. In the Methods section, line 200, we briefly described how the conductance variations were measured: “The resistance of the junctions was measured after each pulse at a low bias of -200 mV (time constant 100 ms). To obtain the STDP curve, we always initialise the junction to the ON state for $\Delta t < 0$ and to an intermediate resistance state between ON and OFF for $\Delta t > 0$. The exact values of initial resistance states in each figure are:

Figure 1d, 3b: $R_{init}(\Delta t < 0) = 6 \times 10^5 \Omega$, $R_{init}(\Delta t > 0) = 1.2 \times 10^7 \Omega$,

Figure 3c: $R_{init} = 1.2 \times 10^7 \Omega$,

Figure 3d: $R_{init}(\Delta t < 0) = 6 \times 10^5 \Omega$, $R_{init}(\Delta t > 0) = 1.2 \times 10^7 \Omega$.”

Concerning the type of the STDP, we can exclude an additive behaviour as the conductance change depends on the initial conductance value (see Figure R1).

Figure R1 : Spike-timing dependent plasticity learning with a ferroelectric synapse : influence of the initial resistance state on the conductance variations. The shape of the pre- and post-synaptic spikes is identical to Figure 3b.

When the initial resistance state is low (e.g. yellow data with $R_{init} = 500\text{-}650 \text{ k}\Omega$), the amplitude of ΔG is maximum for $\Delta t < 0$ while no variations can be detected for $\Delta t > 0$. Reciprocally, when the initial resistance state is high (e.g. orange data with $R_{init} = 9\text{-}15 \text{ M}\Omega$), the amplitude of ΔG is maximum for $\Delta t > 0$ while no variations can be detected for $\Delta t < 0$. Overall, for intermediate initial states of resistance, the shape of the STDP is similar to the data presented in the manuscript.

Furthermore, owing to the slight asymmetric response of the FTJs concerning the polarity of the applied voltage (see, e.g., Figure 3a), we can exclude a strictly multiplicative STDP. Nonetheless, we clearly notice the self-limiting (“self-adaptation”) effect that is expected to be important for the stability of larger neural networks [Rossum et al. *The Journal of Neuroscience* 20, 8812 (2000); Rubin et al. *Physical Review Letters* 86, 364 (2001)]. Finally, as shown in Figure 3b-d, the shape of the STDP curve will however always depend on the exact shape of the pre- and post-synaptic spikes.

#5. Page 7, line 140, "This procedure allows the generation of biologically realistic (Fig. 3b and c) or artificially designed (Fig. 3d) STDP learning curves. Well, the obtained STDP is biologically realistic, except for the time scales involved. Here, the STDP overlap time window is in the order of +/-600ns. In biological synapses, instead of 600ns you would have 60ms to 100ms: 5 orders of magnitude off. This means that your devices could be useful for accelerated neuromorphic systems (assuming the rest of components, besides the synapses, could be made to operate at such acceleration). But, for example, to use them in a robotic self-learning system that operates and interacts with the environment in real time, they could not be used, at least in the way STDP is described in the paper. You should clarify this as well in the paper.

We agree with the Reviewer that the time window we chose for the STDP, 600 ns set by the duration of the ramp in the spikes waveforms, is much faster than in the case of biological synapses. Indeed, we are targeting here high-speed applications useful for, e.g., ultrafast video processing. By choosing much longer durations for the ramp, it should nevertheless be possible to get closer to biological times.

We have modified the sentence line 139 to: “This procedure allows the generation of biologically realistic², though accelerated (Fig. 3b and c), or artificially designed (Fig. 3d) STDP learning curves.”

#6. In page 8, lines 162-165 you mention that recognition performance is critically dependent on spike amplitudes. This highlights the fact that recognition might be also strongly dependent on other parameters as well, and mostly on inter-device parametric mismatch. Since you have a very precise model for the device operation, and have fabricated several devices, there should be a statistical characterization on the variations of the different parameters from device to device. Since the physics is based on ferroelectric phenomena, I would expect (in principle) that the amplitude of spikes is not a critical issue. However, pulse nano-second widths for incremental changes might impact on the mismatch between different devices (this is, two devices may need very different pulse widths to be subject to the same increment/decrement in conductance G). Additionally, there might be other parameters within the devices subject to high inter-device mismatch. Consequently, statistical mismatch characterization is crucial, and I believe the authors have sufficient data to at least provide a Table on this.

Regarding the uniformity of our tunnel devices, the samples studied in the present paper are similar to those presented in Boyn *et al.*, *Appl. Phys. Lett.* 104, 052909 (2014). In this paper we show the distribution of the virgin, OFF, and ON states for 45 working FTJs (out of 50) on the same chip (Figure R2). We note a good reproducibility with clearly separated ON and OFF states from device to device and an average OFF/ON ratio over 10^3 .

Figure R2: Distribution of the virgin, ON and OFF resistance states for 45 working FTJs. The inset shows the distribution of the OFF/ON ratio. Resistance was measured under a DC voltage of -200 mV.

In addition, as illustrated in Figure 2c, the resistance changes in our devices are more sensitive to the amplitude of applied pulses rather than their duration. We do not have enough data on STDP measurements to provide a complete statistical description. However, we can illustrate the device to device reproducibility by the conductance evolutions of three different devices exposed to the same voltage waveform inputs (Figure R3).

Figure R3: a-c, Spike-timing dependent plasticity learning in three different devices (S1, S2, S3, respectively). The pre- and post-synaptic spikes and conductance variations are shown in the top and bottom panels, respectively. The lines are fit results from Eq. (1).

#7. Having high R_{on} (300Kohm) and R_{off} (100Mohm) are a plus for low power. However, in your case, you need also fast dynamics. So, care must be taken with parasitic capacitances. For example, suppose you want to implement a large array. Your STDP dynamics are in the order hundreds of ns. So, you need your currents and spikes to be collected by circuits with parasitic time constants below this, say 10ns. Therefore, the RC time constants of your array lines must be $RC=10ns$ (or less). Let's consider as worst case when one resistance is at min value, about 300Kohm from Fig.3a, and the rest at max 100Mohm. Then $RC<10ns$ implies that $C<30fF$ (neglecting the impact of the 100Mohm resistances). In present day CMOS technologies, long lines tend to have parasitic capacitances in the range of pF. Of course, you plan to have very dense arrays of nano scale synapses, but there will be a limit on scalability determined by your fast STDP time constants, max and min resistances, and line parasitic capacitances. You should include a Section to discuss about what this limit would be.

We agree with the referee that co-integration between CMOS and ferroelectric memristors is a challenge, and this is the next issue that we will address. In the case of a memristive crossbar, the resistance seen by the pre-synaptic neuron is the sum of the memristors in parallel. Therefore, the worst case for the resistance seen by pre-synaptic neurons is when all memristors are simultaneously in the high resistance OFF state ($\sim 100 M\Omega$). On the other hand, for post-synaptic neurons the issue is to deal with high currents, the worst case corresponding to all memristors being simultaneously in the ON state ($\sim 300 k\Omega$). If we integrate for example 500 synapses per post-synaptic neuron, the maximum equivalent resistance is $200 k\Omega$, and the maximum current is of the order of 3 mA. Such current amplitude is compatible with standard CMOS technologies. If we keep the spike waveform presented in the paper, we need to have a time constant at least 10 times smaller than the signal, i.e. about 10 ns. In that case, the equivalent parasitic capacitance is smaller than 50 fF which is also compatible with current CMOS technologies. Increasing the number of synapses decreases the equivalent resistance, allowing a larger parasitic capacitance but at the same time increasing the current that transistors have to deal with. In order to solve these issues, an easy solution will be to modify the shape of the action potentials in order to have smoother transitions and to decrease the constraints in terms of bandwidth.

Reviewers' comments:

Reviewer #1 (Remarks to the Author):

The response of the authors clarifies most of the issues except one for Figure 2e. I agree with the authors that the switching mechanisms for both 10 and 300 K are identical and the inhomogeneous switching governed by Eq. (1) can occur in the T-phase BFO. However, the switching times are different at different temperatures. Even if the switching times computed by the authors agree well with the interpolated ones from experiment results in Figure 2e, the data collected from different temperatures cannot be fitted by a single line. I would recommend reorganizing figures and adding description for the different temperatures.

Reviewer #2 (Remarks to the Author):

This manuscript was at a level where all suggested changes and comments made in my initial report were intended to lead to alterations, only where the authors felt they would enhance the paper. I'm hence very happy to recommend publication of the manuscript on the basis of the changes made.

Reviewer #3 (Remarks to the Author):

Authors have complied with my requests. Congratulations on the nice work.

Reviewers' comments:

Reviewer #1 (Remarks to the Author):

#1. The response of the authors clarifies most of the issues except one for Figure 2e. I agree with the authors that the switching mechanisms for both 10 and 300 K are identical and the inhomogeneous switching governed by Eq. (1) can occur in the T-phase BFO. However, the switching times are different at different temperatures. Even if the switching times computed by the authors agree well with the interpolated ones from experiment results in Figure 2e, the data collected from different temperatures cannot be fitted by a single line. I would recommend reorganizing figures and adding description for the different temperatures.

We thank Reviewer #1 for carefully analysing this manuscript. We agree with the Reviewer that we expect in principle the switching dynamics of the T-phase BFO to change with temperature. The main purpose of MD simulations is to show that nucleation-limited switching in T-phase BFO can be intrinsic. For technical reasons, such simulations are performed at 10 K while experiments are performed at room temperature. Therefore, we reorganized Figure 2e in order to separate experiments from MD simulations. Importantly, in both cases the mean switching time evolves with the electric field according to the Merz's law. We extract an activation field of 3.0 V/nm from experiments and of 2.4 V/nm from MD simulations. Considering the simplifications made during MD simulations as well as the different temperature, we reduced our claim to write that these activation fields are within the same range.

The changes in the manuscript are highlighted in yellow. For Figure 2, we realized that the colour code between Figure 2c, 2d and 2e for the electric field values was not consistent and corrected it.

REVIEWERS' COMMENTS:

Reviewer #1 (Remarks to the Author):

I have reviewed the revised manuscript and I'm happy to recommend publication of the manuscript.